# Use of Essential Oils to Counteract the Phenomena of Antimicrobial Resistance in Livestock Species

**DOI:** 10.3390/antibiotics13020163

**Published:** 2024-02-07

**Authors:** Carmine Lupia, Fabio Castagna, Roberto Bava, Maria Diana Naturale, Ludovica Zicarelli, Mariangela Marrelli, Giancarlo Statti, Bruno Tilocca, Paola Roncada, Domenico Britti, Ernesto Palma

**Affiliations:** 1Mediterranean Ethnobotanical Conservatory, Sersale (CZ), 88054 Catanzaro, Italy; studiolupiacarmine@libero.it (C.L.); fabiocastagna@unicz.it (F.C.); 2National Ethnobotanical Conservatory, Castelluccio Superiore, 85040 Potenza, Italy; 3Department of Health Sciences, University of Catanzaro Magna Græcia, 88100 Catanzaro, Italy; tilocca@unicz.it (B.T.); roncada@unicz.it (P.R.); britti@unicz.it (D.B.); palma@unicz.it (E.P.); 4Ministry of Health, Directorate General for Health Programming, 00144 Rome, Italy; mariadiananaturale@gmail.com; 5Department of Pharmacy, Health and Nutritional Sciences, University of Calabria, Rende, 87036 Cosenza, Italy; ludovicazicarelli@gmail.com (L.Z.); mariangela.marrelli@unical.it (M.M.); g.statti@unical.it (G.S.); 6Center for Pharmacological Research, Food Safety, High Tech and Health (IRC-FSH), University of Catanzaro Magna Græcia, 88100 Catanzaro, Italy

**Keywords:** essential oils, plants, antimicrobial activity, animal husbandry, antimicrobial-resistant bacteria (ARBs), animal health and welfare

## Abstract

Antimicrobial resistance is an increasingly widespread phenomenon that is of particular concern because of the possible consequences in the years to come. The dynamics leading to the resistance of microbial strains are diverse, but certainly include the incorrect use of veterinary drugs both in terms of dosage and timing of administration. Moreover, the drug is often administered in the absence of a diagnosis. Many active ingredients in pharmaceutical formulations are, therefore, losing their efficacy. In this situation, it is imperative to seek alternative treatment solutions. Essential oils are mixtures of compounds with different pharmacological properties. They have been shown to possess the antibacterial, anti-parasitic, antiviral, and regulatory properties of numerous metabolic processes. The abundance of molecules they contain makes it difficult for treated microbial species to develop pharmacological resistance. Given their natural origin, they are environmentally friendly and show little or no toxicity to higher animals. There are several published studies on the use of essential oils as antimicrobials, but the present literature has not been adequately summarized in a manuscript. This review aims to shed light on the results achieved by the scientific community regarding the use of essential oils to treat the main agents of bacterial infection of veterinary interest in livestock. The Google Scholar, PubMed, SciELO, and SCOPUS databases were used for the search and selection of studies. The manuscript aims to lay the foundations for a new strategy of veterinary drug use that is more environmentally friendly and less prone to the emergence of drug resistance phenomena.

## 1. Introduction

One of the biggest issues facing livestock farming across the globe is the fast-growing demand for food of animal origin, particularly in low-income countries. Animal breeding methods are evolving as a result of this expanding tendency, and intensive livestock systems are quickly replacing rural livestock farms [1]. In today’s livestock system, antibiotics play a key role. Particularly, antibiotics are becoming an essential part of animal husbandry in order to fulfill the world population’s demand for food [2,3].

Sick animals may spread disease to other flock members either directly or indirectly via contaminated feed, water, soil, and plants. This is particularly true in intensive livestock, where space for each animal is limited [4]. Furthermore, at any point during the processing of animals in slaughterhouses, germs can be transferred from carcass to carcass [5,6]. Livestock infections have a significant impact on public health as well, and great efforts are made to control them because outbreaks of zoonotic diseases in humans are linked to direct or indirect contact with an infected animal, the animal’s environment, or, more commonly, the consumption of contaminated food of animal origin [7]. Furthermore, antibiotics are widely utilized not just for medicinal purposes but also as growth-promoting, and disease-prevention measures. However, their overuse has increased selection pressure, which has led to the current emergence of antibiotic resistance.

In 2014, the World Health Organization (WHO) stated that the global public health catastrophe posed by antibiotic-resistant bacteria is rapidly approaching [8]. The issue of antimicrobial resistance affects both human and veterinary medicine. Antimicrobial-resistant bacteria (ARBs) infections are on the rise today; it is predicted that by 2050, these infections will be responsible for 10 million deaths per year [9]. ARB infections in livestock are particularly dangerous as they are linked to higher veterinary care costs, higher death rates, declining populations, and decreased production [10].

When treated with antibiotics, animals eliminate the active ingredients or their metabolites into the environment to some extent [11]. The biggest source of environmental pollution is animal dung. After administration, antibiotics are absorbed by the body and processed. The main routes of elimination of the drug and/or its metabolites are urine and/or feces [12]. The animal excretes antibiotics through feces if oral absorption is inadequate.

The presence of antibiotics at levels below those used for therapy accelerates the spread of ARBs. The World Organization for Animal Health (OIE) advised the prudent and cautious use of antimicrobial drugs in animals to reduce this worldwide problem and guarantee the safety of animal-derived food [13]. Antibiotic usage in animal farms has already been reduced in several countries; the European Union (EU) banned the use of antibiotics as growth promoters in animal diets in 2006 [14]. Furthermore, consumer “feeling” is changing. When choosing food, they pay more attention to food labels and often choose products from animal farms where the use of drugs is limited [15,16].

In this scenario, in order to lower ARBs and the frequency of the most significant infections in livestock farming, the search for effective antibiotic alternatives is becoming mandatory [17,18]. Antimicrobial peptides, bacteriophage, bacteriocins, prebiotics, and probiotics have garnered interest as possible replacements or supplements to currently used antimicrobial chemicals [19,20]. To these promising alternatives are added the resources provided by the plant world.

For millennia, traditional medicine has been based on medicinal plants. Through observation and experience, humans have developed a vast traditional culture around the appropriate use of plants. Over the past hundred years or so, advancements in science and technology, particularly in the field of plant pharmacognosy, have led to the identification of several novel compounds with unusual modes of action [21]. The scientific use of drugs obtained from plants for the treatment and prevention of diseases is known as phytotherapy.

The use of phytotherapy in veterinary medicine is also known as ethnoveterinary medicine; the latter is practiced wherever people have a strong connection with animals. It is particularly important in communities where breeding animals are the main source of food. Numerous botanical species have historically been used—and in some cases still are—to cure horses, pigs, sheep, cows, and poultry [22]. In recent decades, ethnoveterinary treatment has become more popular in highly industrialized countries due to public health concerns (such as antibiotic/anthelmintic resistance) and the ongoing expansion of organic livestock farming [23,24]. The primary basis for phytotherapy is derived from natural complexes present in plants, which are mostly composed of low molecular weight chemical compounds. The components of fruits, vegetables, spices, and other plant parts (such as pulp, bark, peel, leaf, berry, or bloom) are sources of phyto-complexes. Today, essential oils (EOs) are among the most widely used natural substances [25]. The culinary, cosmetic, pharmaceutical, and agricultural sectors have all found many uses for EOs, which has led to a current wave of studies on secondary plant metabolites. Thus, studies on the medicinal properties of EOs have increased in recent years [26,27,28,29,30,31].

The investigation of EOs’ antibacterial qualities is becoming more and more important as antibiotic-resistant forms of bacteria pose a serious danger to both human and animal health, making alternative pharmacological treatments useful in the fight against diseases brought on by these pathogens. Numerous studies have examined the effectiveness of EOs both in vivo and in vitro [32,33]. Some of them have assessed the impact of EOs on animals by adding them to feed or sprinkling them on the animals’ quarters to lower bacterial contamination and raise farm cleanliness standards [34]. Other studies have been conducted on reference strains, bacterial strains isolated from farm animals, and food of animal origin to reduce bacterial contamination by extending the shelf life of goods meant for human consumption [35]. Several veterinary studies conducted in vivo and in vitro on the use of EOs as substitutes for available antibiotics were considered in this review. Literature reviews summarizing the various scientific publications in this field are lacking. This manuscript therefore aims to take stock of the situation and infer the suitability of certain compounds as promising candidates for the treatment of microbial diseases.

## 2. Methodology

Several databases were used to gather the papers for this review, including PubMed, Google Scholar, PubMed, SciELO, and SCOPUS. A variety of keywords pertaining to the review’s topic were included in the search box, such as “antibacterial activity”, “phytochemical” or “antimicrobial resistance”, and “essential oil and antibacterial activity.” Depending on the combination of phrases, the operators “AND” or “OR” were employed. Using “antibacterial activity” AND “plant extract” OR “essential oil” as examples. To be eligible for the evaluation process, the studies had to be published in English. Research on particular antimicrobial substances was limited to those with detailed descriptions of their mechanisms of action, accurate quantitative assessments of their antibacterial efficacy, and a well-established origin.

## 3. Essential Oils

The European Pharmacopoeia defines EO as: “Odorant product, generally of a complex composition, obtained from a botanically defined plant raw material, either by driving by steam of water, either by dry distillation, or by a suitable mechanical method without heating” [36]. According to the European Pharmacopoeia, an EO is typically extracted from the aqueous phase using a physical process that does not significantly alter its chemical composition. Due to their strong flavor, which decreases the appetite of herbivores, EOs have a significant protective effect on plants. When in the liquid phase, EOs are volatile, limpid, and seldom colored substances. Chemically, they are soluble in organic solvents that have lower densities than water [37]. All plant parts, including buds, flowers, leaves, stems, twigs, seeds, fruits, roots, wood, and bark, are useful to synthesize. Once generated, EOs are stored in glandular trichomes, secretory cells, cavities, and canals [38]. Only 10% of plant species—more than 17,000 plant species—produce and contain EOs, consequently, they are referred to as aromatic plants. The Lamiaceae, Lauraceae, Asteraceae, Rutaceae, Myrtaceae, Poaceae, Cupressaceae, and Piperaceae families make up the majority of their global distribution [39].

EOs are made up of a naturally occurring complex combination produced by plants’ secondary metabolism. This mixture typically contains 20–60 organic volatile components with molecular weights below 300. They are partially found in the vapor form because of their high enough vapor pressure at ambient temperature and atmospheric pressure [40]. Two or three primary constituents, present in high concentrations (20–70%), make up each EO. Terpenes (including monoterpenes and sesquerpenes), aromatic chemicals (such as alcohol, phenol, methoxy derivatives, and aldehydes), and terpenoids (isoprenoids) are generally the principal active ingredients. Terpene hydrocarbons and oxygenated molecules are the two main categories into which EO components and aroma can be separated [38,41]. The most prevalent class of chemical components included in EOs are terpenes and terpenoids. When several isoprene units (C5H8) are combined, terpenes, which are hydrocarbons, are produced. The production of terpenes takes place in the cytoplasm of plant cells, is initiated by acetyl-CoA, and progresses via the mevalonic acid pathway. Cyclases, which have a hydrocarbon backbone, are able to reorganize terpenes into cyclic structures, resulting in the formation of either monocyclic or bicyclic structures [42,43]. Terpene-specific synthetase modifies the allylic prenyldiphosphate to form the terpene skeleton. After synthesizing the isopentenyl diphosphate (IPP) precursor, IPPs are repeatedly added to form the prenyldiphosphate precursor of the various classes of terpenes. Finally, secondary enzymatic modification (redox reaction) of the skeleton confers functional properties to the various terpenes [42,43]. The two primary types of terpenes are monoterpenes (C10H16) and sesquiterpenes (C15H24), however, there are also longer chains like diterpenes (C20H32), triterpenes (C30H40), etc. Some examples of EO components having antimicrobial action include p-cymene, limonene, menthol, eugenol, anethole, estragole, geraniol, thymol, γ-terpinene, and cinnamonyl alcohol [44]. Plants that contain some of these chemicals include angelica, bergamot, lemongrass, mandarin, mint, caraway, celery, citronella, coriander, eucalyptus, geranium, petitgrain, pine, juniper, lavandin, lavander, lemon, orange, peppermint, rosemary, sage, and thyme [45].

Terpenoids are produced by biochemically modifying terpenes using enzymes that transfer or remove methyl groups and add oxygen molecules [42]. Alcohols, phenols, esters, aldehydes, ethers, ketones, and epoxides are the different subgroups of terpenoids. Terpenoids include thymol, carvacrol, linalool, linalyl acetate, citronellal, piperitone, menthol, and geraniol.

Phenylpropanoids (derived from phenyl-propane) are aromatic compounds produced via the shikimic acid route that leads to phenylalanine. The aromatic chemicals include phenols (eugenol and chavicol), aldehydes (cinnamaldehyde), alcohols (cinnamic alcohol), methylenedioxy compounds (apiole, myristicin, and safrole), and methoxy derivatives (anethole, estragole, and methyleugenols).

EOs exhibit a relatively high degree of variety in terms of qualitative and quantitative composition.

This variability is caused by a number of variables, which fall into two main categories: (1) intrinsic factors, which include the plant itself, its maturity, and its interactions with the environment (such as the type of soil and climate); and (2) extrinsic factors, which are associated with the extraction technology.

Seasonal fluctuations, plant organs, plant maturity, regional origin, and genetics are other aspects to consider. Due to their strong interdependence and mutual effect, these aspects can sometimes be challenging to separate from one another [40].

### Antimicrobial Mechanism of Action

The composition of EOs, the presence of functional groups in their active components, and their synergistic interactions are factors that determine their activity [46]. The kind of EO or strain of the microbe utilized determines the antibacterial mode of action. It is well recognized that Gram-positive bacteria are more vulnerable to EOs than Gram-negative bacteria [47,48]. This is explained by the fact that Gram-negative bacteria have an outer membrane that is more complex, rigid, and rich in lipopolysaccharides (LPS), which limits the diffusion of hydrophobic compounds through it. In contrast, Gram-positive bacteria lack this extra complex membrane and are instead surrounded by a thick peptidoglycan wall that is not dense enough to withstand small antimicrobial molecules, which facilitates the entry of antimicrobial compounds through the cell membrane. Moreover, since lipophilic ends of lipoteichoic acid are present in cell membranes, the membrane composition of Gram-positive bacteria may facilitate the penetration of hydrophobic components of EOs [49].

Numerous studies have shown that the bioactive ingredients found in EOs can adhere to the cell’s surface before penetrating the phospholipid bilayer of the cell membrane. Their buildup compromises the structural integrity of the cell membrane, which can negatively impact cell metabolism and ultimately result in cell death [50,51]. Additionally, it has been reported that the action of EOs on the integrity of the cell membrane alters its permeability, causing the loss of essential intracellular components such as proteins, reducing sugars, ATP, and DNA, while also inhibiting the generation of energy (ATP) and related enzymes, resulting in cell breakdown and electrolyte leakage [52]. Therefore, a series of events encompassing the whole bacterial cell is thought to be responsible for the antimicrobial activity of EOs [53]. Figure 1 summarizes the mechanisms of toxic action of EOs against the bacterial cell.

Table 1 shows some EOs and their components that have demonstrated antimicrobial action. The mechanisms by which they exert this action on the bacterial cell are also mentioned.

## 4. Control of Bacterial Disease via Essential Oils

Etiological agents of infection that can affect a large number of animals are common in intensive livestock farms; the most frequently affected animals are the young ones [61,62]. Equally frequent are infectious diseases that can affect individual animals. It is often opportunistic bacteria that sustain these pathological states, often conditioned by alterations in normal physiological conditions, as occurs during pregnancy. Opportunistic bacteria have become particularly important because the evolution of animal husbandry technologies often leads to a drop in the host’s protective reactivity, such that it is vulnerable even to bacteria proper to the normoergic host [63]. In general, several bacterial strains are developing resistance to synthetic drugs. Table 2 shows some important infectious agents and the main classes of antibiotics against which resistance has been recorded. In this situation, natural products, in particular Eos, could be a viable alternative. The results obtained by the scientific community in controlling the etiological agents of important bacterial diseases using EOs isolated from different botanical species are examined in the next paragraphs.

### 4.1. Gram-Negative Bacteria

#### 4.1.1. *Escherichia coli*

*Escherichia coli* is a member of the Enterobacterales family responsible for numerous diseases in both humans and animals. It poses a serious risk to veterinary care as it can spread disease to all animal species, often resulting in financial losses. In calves, it is responsible for colibacillosis. This is an infection that can occur from the first days of life until 2–3 months of age. *E. coli* infects the calf soon after birth through the food route or the umbilical cord and colonizes the intestine. Depending on the age of the subjects, the serotypes involved, and their ability to settle in the various intestinal tracts, septicemic, enterotoxemic, and enteric forms of colibacillosis can occur [75]. In cattle, *E. coli*, *Klebsiella*, *Enterobacter*, and other microorganisms are responsible for mastitis [76]. Furthermore, infection by *E. coli* can result in avian colibacillosis, a systemic disease affecting birds. While contaminated dust is typically inhaled by animals, *E. coli* mostly affects poultry via an oral–fecal cycle. It produces a wide range of clinical symptoms in chickens, including enteritis, cellulitis, enlarged head syndrome, septicemia, polyserositis, coligranuloma, omphalitis, and salpingitis. Poultry colibacillosis is characterized by subacute symptoms such as pericarditis, airsacculitis, and perihepatitis, and acute symptoms such as septicemia that can be fatal [77].

Septicemia is often the first sign of the infection, followed by localized inflammation in many organs or abrupt death [78]. Avian colibacillosis is characterized by variable lesions, the most common of which are polyserositis and airsacculitis. Furthermore, *E. coli* can penetrate the egg shell if the eggs are contaminated with feces. Contamination of eggs by feces can cause yolk sac infection or allow *E. coli* to enter through the shell, infecting the chicks during hatching and frequently leading to significant mortality rates. When infected at an early stage, birds that survive clinical disease may not exhibit many symptoms of infection but may become carriers [79]. Economic losses due to necrotic cellulitis may occur at slaughter. Given the importance of this disease, several studies were conducted to confirm the anti-*E. coli* properties of some EOs derived from various botanical species. The goal of these studies was to identify natural products that could be used to improve environmental hygiene on chicken farms. Several EOs showed anti-*E. coli* action in vitro.

Rhayour et al. (2003) discovered that clove EO had an impact on *E. coli*, causing holes in the membrane and cell wall [80].

A study was carried out by Burt & Reinders (2003) [81] to measure the bactericidal qualities of five different EOs on a non-toxic strain of *E. coli* O157:H7 at three different temperatures and with and without an emulsifier or stabilizer. The disc diffusion test was used to screen for antibacterial qualities, and the most active ones were chosen for additional investigation using microdilution colorimetric assays. The greatest bacteriostatic and bactericidal effects were exhibited by *Origanum vulgare* and *Thymus vulgaris* (light and red variants) EOs, followed by *Syzygium aromaticum* and *Pimenta racemosa* EOs. Sixty-five μL L^−1^ of *Origanum vulgare* EO was colicidal at 10, 20, and 37 °C. While 0.25% (*w*/*v*) lecithin decreased bacterial activity, the inclusion of 0.05% (*w*/*v*) agar as a stabilizer strengthened the antibacterial capabilities, especially at 10 °C [81].

Applied alone or in combination, *Cinnamomum verum* and *Syzygium aromaticum* EOs have demonstrated strong antibacterial action against *E. coli* strains obtained from chickens with colibacillosis.

The primary components of the two oils—eugenol and its acetate form—have been linked to this action [82]. According to Zhang et al. (2016) [83], *Cinnamomum verum* EO alters the *E. coli* membrane’s permeability and integrity, resulting in the loss of inner cell components [83].

EOs extracted from the leaves of 29 medicinal plants in Brazil were tested in a study by Duarte et al. (2007) [84] against 13 distinct *E. coli* serotypes. The EO derived from *Cymbopogon martini* demonstrated a wide range of inhibition and high action (MIC between 100 and 500 μg/mL) against ten of the thirteen *E. coli* serotypes, including two enteropathogenics, three enteroinvasives, three enterotoxigenics, and two shiga-toxin-producing serotypes. Two enterotoxigenic, one enteropathogenic, one enteroinvasive, and one sigatoxin-producing serotype were all significantly suppressed by *Cymbopogon winterianus*. *E. coli* can be effectively killed by *Aloysia citrodora* with moderate to severe inhibition. Other EOs exhibited antibacterial qualities as well, although their effectiveness against the investigated serotypes was more constrained [84].

Raho and Benali (2012) [85] investigated the antibacterial properties of the EO derived from *Eucalyptus globulus* leaves in vitro. The EO’s inhibitory properties were evaluated using dilution broth and agar disc diffusion techniques against *E. coli*. The findings demonstrated that, depending on the inoculum size and the quantity of EO, the EO exhibited antibacterial activity to differing degrees. The EO of *Eucalyptus globulus* leaves has an inhibitory zone ranging in diameter from 8 to 26 mm. The largest zone of inhibition was obtained for *E. coli* (10^−3^ dilution) with a 100% concentration of *Eucalyptus globulus* EO.

Romero et al. (2015) [86] investigated the antibacterial impact and mechanism of action of some components of EOs (carveol, carvone, citronellol, and citronellal) against *S. aureus* and *E. coli*. Dimethyl sulfoxide (DMSO, 100%) at different concentrations ranging from 0.066 to 3000 μg/mL was used to test each component. The compounds were weighed to obtain μg/mL units. The microdilution broth technique was used to calculate the MICs of the EO components. When tested against *E. coli*, every chemical showed bactericidal and inhibitory properties. Citronellol (5 μg/mL) proved highly inhibitory against *E. coli*, with carveol/carvone (200 μg/mL) and citronellal (300 μg/mL) following [86].

The study by Thielmann et al. (2019) [87] is one of the most significant in terms of the quantity of plant species tested against *E. coli*. Using a single-method approach, the authors determined the minimum inhibitory concentrations (MICs) of 179 EO samples from 86 plant species. Using a dispersion-based method to integrate EOs in a concentration range between 6400 and 50 μg/mL, MICs were obtained in a broth microdilution experiment. Twenty-two samples from 12 plant genera inhibited *E. coli* at doses below 400 μg/mL.

At 50 μg/mL, only four EOs—*Neolitsea cassia*, *Cinnamomum verum*, *Origanum vulgare*, and *Thymus zygis*—showed efficacy against *E. coli*. Despite this, certain additional EOs from *Cupressus sempervirens*, *Backhousia citriodora*, *Cymbopogon citratus*, *Cymbopogon martini*, *Cymbopogon nardus*, *Litsea cubeba*, *Origanum majorana*, *Origanum vulgare*, and *Syzygium aromaticum* nevertheless showed encouraging action against *E. coli*, with MICs ranging from 100 to 400 μg/mL. Some EOs did not show antibacterial properties. In particular, *Cananga odorata*, *Cupressus sempervirens*, *Daucus carota*, *Foeniculum vulgare*, *Juniperus communis*, *and Pimpinella anisum* had no efficacy against *E. coli* [87].

Minimum inhibitory concentrations (MICs) and minimum bactericidal concentrations (MBCs) were used in the study by Galgano et al. (2022) [88] to test the bacteriostatic–bactericidal activity of *Citrus limon*, *Pinus sylvestris*, *Foeniculum vulgaris*, *Ocimum basilicum*, *Melissa officinalis*, *Thymus vulgaris*, and *Zingiber officinale* against various *E. coli* strains in vitro. Except for *T. vulgaris*, all the studied EOs showed little antibacterial action against *Escherichia coli*. The antibacterial action of *T. vulgaris* was independent of oil content, and this EO completely inhibited the growth of *E. coli* [88].

It has also been shown that Litsea cubeba EO works well against *E. coli* [89]. When *E. coli* cells were treated with this EO, Li et al. (2014) [90] noticed holes and gaps on the outer and inner membranes. These were mostly ascribed to the existence of aldehydes like geranial and neral.

The strong action against *E. coli* isolates that the EO of *Cymbopogon citratus* showed in vitro can be traced back to aldehydes [89,91].

High action against *E. coli* was also shown by *Mentha piperita* EO [89,92]. The main ingredients of *Mentha piperita* EO, menthol, and its oxidative product, menthone, have been shown to possess antibacterial activities [93,94].

Eos from *Pelargonium graveolens* and *Ocimum basilicum* showed anti-*E. coli* action [89,95,96]. The significant quantity of linalool in basil EO has been linked to its activity, but to a varied degree depending on the strain tested, whereas the high concentration of oxygenated monoterpenes in *Geranium robertianum* EO seems to influence its antibacterial function [89].

#### 4.1.2. *Salmonella* Species

Pathogenic to humans and animals, *Salmonella* spp. Causes gastroenteritis, septicaemia, pneumonia, and abortion. Some serovars, referred to as “host-restricted”, are closely adapted to the host; for example, *S. abortusovis* is adapted to sheep and is the main cause of abortion in this animal, while *S. typhi suis*, *S. gallinarum*, and *S. pullorum* are adapted to pigs and chickens, respectively. Other serovars possess a broad host spectrum and are known as “host-adapted” serovars. Diseases in pets and livestock animals result in financial losses and increase the possibility of transmission to people. In people, infection can occur through the ingestion of contaminated food and drink. The field of food microbiology has mostly studied the application of EOs in combating *Salmonella* infections. Indeed, foodborne diseases caused by *Salmonella* spp. pose a growing risk to human health; salmonellosis outbreaks are the result of recent changes in food processing and marketing practices [97]. Relatively few serovars are responsible for the majority of cases of salmonellosis in people and domestic animals; these can be divided into three categories based on host predominance.

The first group of serovars is host-specific. They usually cause systemic diseases in a few animals that are related by phylogeny. For instance, systemic disease in sheep, poultry, and humans is nearly solely linked to *S. enterica* serovar *abortusovis*, *paratyphi*, and *pullorum*. The second type of strain is host-restricted. Although they are mostly associated with one or two closely related host species, they can typically lead to disease in different hosts. For example, serovar Dublin and serovar Choleraesuis of *S. enterica* are often linked to severe systemic diseases in ruminants and pigs, respectively [98]. However, it’s possible for these serovars to infect humans and other animal species. The third category is *S. enterica* serovars, including *infantis* and *enteritidis*, which often cause gastroenteritis in a significant number of unrelated host species. Of course, there are wide variations in the type and severity of *Salmonella* infections in various animal species. These variations are caused by a variety of factors, such as the *Salmonella* serovar, dosage, age, virulence of the strain, host animal species, host immunological state, and geographic location [99]. Worldwide, *S. enterica* continues to be a major source of infection and disease in both humans and animals.

The R (+)-limonene, orange terpenes, trans-cinnamaldehyde, carvacrol, and cold compressed orange oil were evaluated against fifteen distinct strains of *S. enterica* subsp. serovar Heidelberg, recovered from a variety of sources, including cattle, pigs, turkeys, poultry, and poultry egg houses. The growth of all microorganisms was entirely suppressed by carvacrol and trans-cinnamaldehyde. However, cold compressed orange oil only hindered the development of two isolates, while R (+)-limonene and orange terpenes exhibited no inhibitory effect against the same strains [100].

EOs extracted from *Aloysia triphylla*, *Cinnamomum verum*, *Cymbopogon citratus*, *Litsea cubeba*, *Mentha piperita*, and *Syzygium aromaticum* have been put to the test with *S. enterica* Serovar Enteritidis isolated from chickens. The anti-*Salmonella* activity of *Cinnamomum verum* was greatest, followed by *Syzygium aromaticum* extract. It has been proposed that the inclusion of these EOs into chicken diets, together with *Saccharomyces cerevisiae* administration, is an integrated strategy to prevent intestinal colonization by *Salmonella*. Additionally, EOs isolated from *Cinnamomum verum* and *Syzygium aromaticum* could be used either alone or in combination for disinfection [101].

The primary ingredients in these EOs, cinnamonaldehyde and eugenol, have been linked to their antibacterial properties. Because of their capacity to donate hydrogen, these compounds react with lipid and hydroxyl radicals to transform them into stable products [102]. Furthermore, since the two substances include a carbonyl group that binds to the enzymes, rendering them inactive and/or damaging the bacterium’s cell wall, they can prevent bacteria from producing vital enzymes [103].

Research against *S. typhimurium* ATCC 14028 revealed notable seasonal variations in the EO components of *Aristolochia fontanesii* roots and their antibacterial action. The main components of *Aristolochia fontanesii* EO were oxygenated monoterpenes (5.9–28.0%) and oxygenated sesquiterpenes (50.2–81.1%). Because of the higher concentrations of alcohols and esters (23.0% of total oil) and the synergistic effect of other minor compounds such as linalool, terpinene-4-ol, camphor, and t-pinocarveol, the EO (15 μL) of *Aristolochia fontanesii* exhibits potent antibacterial (zone of inhibition: 8.2–12.0 mm) properties [104]. Nearly all tested *Salmonella* strains were shown to be significantly susceptible to the EOs of *Thymus vulgaris*, *Syzygium aromaticum*, and *Cinnamomum verum*. The clove, cinnamon, and thyme Eos’ MBC/MIC ratios for most bacteria were equal to 1, indicating that these oils had bactericidal effects at concentrations of at least 20 μL/mL. Microbes’ ability to form biofilms was seen to be suppressed in the presence of subinhibitory concentrations of *Thymus vulgaris*, *Cinnamomum verum*, and *Syzygium aromaticum* EOs [105]. The oxygenated monoterpene concentration of EOs isolated from *Thymus mastichina* and *Calamintha nepeta* was greater (86.0–91.0%), with 1,8-cineole (28.0–71.0%) as the most important component. On the other hand, *Origanum virens* EO exhibited a greater concentration of hydrocarbon monoterpenes (45.3%) and oxygenated monoterpenes (45.5%), with notable components including γ-terpinene (20.2%) and thymol (19.4%). When compared to other tested Gram-negative strains in the same investigation, *S. enteritidis* LFG 1005 and *S. typhimurium* LFG 1006 were more vulnerable to the strong antibacterial activity of the EOs of all three of these plants. The high content of 1,8-cineole in the EOs of *Calamintha nepeta* and *Thymus mastichina* was shown to exhibit strong antibacterial activity, with MICs ranging from 0.8 to >2 mg/mL [106].

In the EO isolated from the leaves of *Taxodium distichum*, 37 chemicals were found. Of all 37 compounds, α-pinene was identified as the main constituent with 83.1% [107]. With an inhibition zone of 17 mm and a minimum inhibitory concentration (MIC) of 78.1 g/mL, *S. typhimurium* was shown to be sensitive to *Taxodium distichum* leaf EO (ATCC 14028). According to Al-Sayed (2018) [107], the presence of terpene components in the EOs is responsible for the reported antibacterial action. In another investigation, it was found that *Spondias pinnata* leaves and flowers contain high levels of β-caryophyllene, a sesquiterpene hydrocarbon, with 49.9% in the leaves and 53.3% in the flowers, while nonacosane (25.0%) was a common ingredient in the EO extracted from the fruit. While the EO from flowers showed poor activity, the EO from leaves demonstrates considerable antibacterial action against *S. typhymurium* (ATCC 14028). Against *S. typhimurium*, leaf EO had a minimum inhibitory concentration (MIC) of 125.0 μg/mL, while gentamycin had a MIC of 0.5 μg/mL. The compounds shown to have antibacterial action against *S. typhimurium* were β-caryophyllene, α-terpineol, α-pinene, β-pinene, terpinolene, selinene, and limonene.

A further investigation revealed that the EO isolated from the leaves of *Artemisia dracunculus* var. *qinghaiensis* had moderate inhibitory effects at a concentration of 10 μL/mL against *S. paratyphi* (CMCCB 50094), with inhibition zones spanning from 10.03 to 13.03 mm [108]. Among the substances found in *Artemisia dracunculus* EO were α-pinene, β-phellandrene, linalool, terpinen-4-ol, α-terpineol, β-caryophyllene, germacrene D, and caryophyllene oxide, which were also identified as antibacterial substances from other plants. Overall, *Artemisia dracunculus*’s high content (19.2%) of sabinene has been linked to its antibacterial action against *S. paratyphi* [108]. Carvacrol (825.0–950.0 μg/mg) was found to be the main constituent of the EO isolated from *Satureja montana* (Lamiaceae) in another study. *Satureja montana* was found to have an in vitro minimum inhibitory concentration (MIC) of 250 μg/mL against *Salmonella enterica* sv Anatum SF2 [109]. According to this study, hydrophilic lipopolysaccharides (LPS), which function as a barrier to macromolecules and hydrophobic chemicals, preventing them from penetrating the target cell membrane, are thought to be responsible for Gram-negative bacteria’s lower susceptibility to EOs. Additionally, in another study, 28 distinct EOs, isolated from different parts of plants, were tested for their antibacterial potential against *L. innocua* ATCC 33090 and *S. enteritidis* ATCC 13076. The most effective EOs against *S. enteritidis* were found in the leaves of *Eucalyptus globulus*, *Eucalyptus exserta*, and *Pimenta pseudocaryophyllus*, as well as two EOs isolated from orange juice by-products *(Citrus sinensis)*: orange oil phase essence and citrus terpenes [110].

#### 4.1.3. *Klebsiella* Species

Environmental sources of *Klebsiella* bacteria include soil and water [111]. *K. pneumoniae* is regarded as one of the most hazardous multidrug-resistant bacteria [112], and is more common in insects, domestic animals, and wild animals than in the environment [113]. It can cause septicemia in foals, pneumonia, metritis, mastitis, and cervicitis in mares [114]. It can lead to lower milk production, worse milk quality, and increased death rates in cows. It can also be the causative agent of mastitis and pneumonia in cows [115]. Strains of *K. pneumoniae* that produce carbapenemase have been described in several papers during the past 20 years. The genesis of multidrug-resistant *K. pneumoniae* phenotypes is caused by the synthesis of these extended-spectrum β-lactamases and modification of the outer membrane permeability (decreased expression of porins and overexpression of efflux pumps) [116]. Since carbapenemases are able to hydrolyze practically all β-lactam antibiotics and even β-lactamase inhibitors, only a small number of second-line antibiotics, either alone or in combination, such as colistin, tigecycline, fosfomycin, polymyxin B, and gentamicin, can still be effective against carbapenem-resistant *K. pneumoniae* [117,118]. Cases of colistin-resistant *K. pneumoniae* have been reported [119]. When choosing second-line anti-*K. pneumoniae* drugs, specific toxicity aspects should be taken into account (gentamicin can potentially cause nephrotoxicity, and colistin is known to have nephrotoxic and neurotoxic effects). For these reasons, the search for alternatives is mandatory. Several plant species have shown antibacterial activity against *K. pneumonia*: *Arbutus unedo* [120]; *Cistus* spp. [121]; *Cytinus hypocistis* [122]; *Myrtus communis* [123]; *Pistacia lentiscus* [124]; *Teucrium* spp. [125]; *Cytinus* [126]; *Thymus vulgaris* L. [127]; *Pistacia terebinthus* [128]; *Rapa catozza* [129]; *Crinum angustum* [130]; *Tinospora cordifolia*, *Alstonia scholaris* [131]; *Rhus coriaria* [132]; *Calicotome villosa* [133]; *Melaleuca alternifolia* [134]; *Mentha* spp. [135]. Among these, *Pistacia lentiscus* [124], *Myrtus communis* [123], and *Arbutus unedo* [120] are undoubtedly common and readily available plants that could be utilized to enhance animal welfare and help fight this infection [136].

It has been shown that several EOs and volatile chemicals make clinical isolates or reference strains of *K. pneumoniae* more susceptible to conventional antibiotics. Research examining the synergistic effects of two native Moroccan thymes (*Thymus maroccanus* and *Thymus broussonetii*) EOs in conjunction with conventional antibiotics against a clinical isolate of *K. pneumoniae* was conducted. While *Thymus broussonetii* EO only exhibited synergistic effects with pristinamycin, *Thymus maroccanus* EO demonstrated synergistic effects with ciprofloxacin, gentamicin, and pristinamycin. The different chemical compositions of these two EOs may be the cause of their differing behaviors: carvacrol content was greater in *Thymus maroccanus* EO (76.35%) than in *Thymus broussonetii* EO (39.77%) [137]. *Thymus saturejoides* Coss. EOs, which have carvacrol as their primary ingredient (25.3–45.3%), worked in concert with cefixime to counteract a clinical isolate of *K. pneumoniae* [138]. The various targets on which the combinations act, including the bacterial membrane (cefixime, carvacrol), proteins (gentamicin, pristinamycin), enzymes (carvacrol), ATP (carvacrol), and DNA (carvacrol, ciprofloxacin), explain these synergistic effects [139,140,141]. *Satureja kitaibelii* EO has been shown to effectively work in concert with two conventional antibiotics, tetracycline and chloramphenicol, to combat *K. pneumoniae* ATCC 700603 (10-fold decrease in the MIC values of both drugs). Savory EO’s main ingredient, geraniol (50.4%), showed synergistic benefits only when combined with chloramphenicol (10-fold reduction in the MIC value), demonstrating antibacterial activity comparable to that of savory EO. By acting on the efflux pumps responsible for antibiotic resistance, geraniol modulates bacterial resistance [142,143]. The combination of geraniol and tetracycline has shown additive benefits; it appears that other chemicals in savory EO (limonene, ocimene, linalool, nerol, β-caryophyllene, and germacrene D) may also work in concert with tetracycline [142]. Against the same reference strain, *K. pneumoniae* ATCC 700603, *Thymus pulegioides* EO, which belongs to the geraniol/geranyl acetate chemotype, has shown synergistic action with tetracycline, streptomycin, and chloramphenicol [144].

#### 4.1.4. *Pseudomonas* Species

*Pseudomonas aeruginosa* is a frequent source of infections in people, animals, and hospitalized patients. In animals, the morbid forms resulting from infection take the form of generalized septicemic-type phenomena or inflammatory processes of a predominantly purulent nature, circumscribed to various organs and apparatuses. It can cause skin, genital tract, and urinary tract infections, as well as bacteremia and pneumonia. Although less common, other species of the *Pseudomonas* genus can infect various body districts. The disease causes widespread deaths of embryos and significant mortality rates in newborn chickens. Fish with *P. aeruginosa* infection exhibit severe symptoms such as hemorrhagic septicemia, gill necrosis, distended abdomen, splenomegaly, friable liver, and enlarged kidney.

*P. aeruginosa* has become resistant to several drug classes. The pathogen’s outer membrane, which shields it from harmful substances and enables it to build a biofilm, is specifically responsible for the bacterium’s innate resistance to a broad spectrum of antibiotics. The inefficacy of some antibiotics against *P. aeruginosa* can also be attributed to the presence of extended-spectrum β-lactamases in this microbe [145,146]. To fight this infection, more than 23 distinct plants can be employed.

Among these plants are *Citrus* spp., which are widely cultivated in the southern region of the Italian peninsula.

EOs extracted from *Cinnamomum* spp. bark have been shown to have antibacterial action against several microorganisms, including *P. aeruginosa* [32,147].

According to research by Kavanaugh et al. [148], cinnamon EO can prevent *P. aeruginosa* from growing and forming biofilm. *P. aeruginosa* was effectively counteracted in vitro by the oils of *Cinnamomum aromaticum*, *Syzygium aromaticum*, *Myroxylon balsamum*, *Thymus vulgaris*, and *Melaleuca alternifolia*. Additionally, the EOs of *Cinnamomum aromaticum*, *Thymus vulgaris*, and *Myroxylon balsamum* were found to be effective in inhibiting biofilm development.

In previous investigations, *T. vulgaris* EO proved effective against *P. aeruginosa*, as it impeded the in vitro proliferation of human clinical multidrug-resistant isolates [149].

But thyme EOs weren’t always effective against this infection. When a *P. aeruginosa* strain was isolated from a dog that had externa otitis, for instance, tests revealed that it was resistant to many antibiotics, just as it was resistant to *T. vulgaris* oil. The canine strain was further tested using oils extracted from the following plants: *Origanum basilicum*, *Origanum vulgare* subsp. *hirtum*, *Litsea cubeba*, *Lavandula latifolia*, *Illicium verum*, *Rosmarinus officinalis*, *Salvia sclarea*, *Ocimum basilicum*, and *Rosmarinum officinalis*. Only the last three oils had little efficacy [150].

Kacaniova et al. (2017) [151] conducted an investigation to explore the EOs qualities towards *Pseudomonas* spp. isolated from fish. A variety of EOs were tested for their antibacterial efficacy against ten distinct strains of *Pseudomonas* isolated from recently captured freshwater fish. The following 21 EOs were used to test each isolate: *Lavandula angustifolia*, *Cinnamomum verum*, *Pinus cembra*, *Mentha piperita*, *Foeniculum vulgare*, *Pinus sylvestris*, *Satureja hortensis*, *Origanum vulgare*, *Pimpinella anisum*, *Rosmarinus officinalis*, *Salvia officinalis*, *Abies alba*, *Citrus aurantium* var. dulce, *Citrus sinensis*, *Cymbopogon nardus*, *Mentha spicata* var. crispa, *Thymus vulgaris*, *Carum carvi*, *Thymus serpyllum*, *Ocimum basilicum*, and *Coriandrum sativum*. Every tested EO showed antibacterial activity. The most successful EO against *Pseudomonas* species was *Cinnamomum verum*, both in accordance with the MIC and disc diffusion techniques. Hosseiny et al. verified the eventual synergism of EOs and antibiotics. The authors used the disc diffusion assay to assess the antibacterial activity of three EOs (*Thymus vulgaris*, *Origanum majorana*, and *Salvia officinalis*) against *P. aeruginosa* (ATCC 9027), either alone or in combination with conventional antibiotics (piperacillin, cefepime, meropenem, gentamicin, and norfloxacin). The examined EOs demonstrated a range of actions against the tested microorganisms, according to the results. Sage showed little effect against the tested microorganisms, whereas thyme and marjoram oils were the most effective. The antibacterial activity of antibiotics that target the cell wall, such as cefepime, meropenem, and piperacillin, was more than doubled when thyme oil was added. The action of every antibiotic studied, with the exception of norfloxacin, was enhanced by marjoram oil. Sage was used with meropenem, gentamicin, and piperacillin to increase their combined action, even though it was ineffective against *Pseudomonas*.

#### 4.1.5. *Campylobacter* Species

Pathogens belonging to the genus *Campylobacter* cause illnesses in humans and several animal species. The *Campylobacter* genus therefore includes micro-organisms responsible for contagious infectious diseases. Domestic animals are sensitive, with symptoms affecting the genital system (abortion, metritis, and sterility), the digestive system (enteritis and hepatitis), and the mammary system (mastitis). The animal infection is transmissible to humans. It is regarded as one of the main causes of gastroenteritis in people globally. *Campylobacter* is the etiological agent of outbreaks linked to chicken products and can cause foodborne disease through the contamination of poultry carcasses.

*Campylobacter* was the cause of 1087 of the 4598 hospitalizations due to foodborne illness in 2015 (Centers for Disease Control and Prevention [CDC], 2017) [152]. Among bacterial infections, this impact on public health was surpassed only by that of *Salmonella* [153,154]. Averaging the number of deaths and illnesses that resulted in health loss, Scallan et al. (2015) [154] calculated that *Campylobacter* caused 22,500 disability-adjusted life years (DALY) every year. In terms of the most significant impact on public health resulting from foodborne illnesses, *Campylobacter* ranked third, behind *Toxoplasma* (32,700 DALY) and non-typhoidal *Salmonella* (32,900 DALY). After rotavirus, typhoid fever, and cryptosporidiosis, a comprehensive review conducted between 1990 and 2013 revealed *Campylobacter* as the fourth most common cause of diarrheal illness [155].

It has been repeatedly shown that broiler and layer flocks have very high *Campylobacter* prevalence. Since EOs seem to strengthen poultry’s immune systems, there is a chance to lower *Campylobacter* concentrations via immune system regulation [156]. Moreover, EOs may directly interact with populations of *Campylobacter*.

Several studies have shown the in vitro activity of various EOs against *Campylobacter* strains.

The effectiveness of EOs against *Campylobacter* isolates is connected to the EO, since each has a distinct chemical makeup and mode of action.

Friedman et al. (2002) [157] evaluated the antibacterial efficacy of many Eos using isolates from clinical and food sources. The most effective Eos against *C. jejuni* were *Citrus aurantium*, *Daucus carota*, *Apium graveolens* seed, *Artemisia vulgare*, *Nardostachys jatamansi*, *Gardenia jasminoides*, *Jasminum abyssinicum*, *Pogostemon cablin*, and *Calendula officinalis.*

It was shown that the EOs of *Melaleuca alternifolia*, *Backhousia citriodora*, and *Leptospermum scoparium* were effective against *C. jejuni* and *C. coli* isolated from chicken [158], while *Origanum syriacum* was effective against a strain of *C. jejuni* [159].

Also, when applied as a feed supplement, the EOs demonstrate good antibacterial activity. When added to diet in quantities less than 1%, caprylic acid, a component of coconut oil and palm kernel oil, greatly decreased the amounts of *Campylobacter* [160,161]. This was noted in 10-day-old and market-age broilers; it had no effect on body weight or feed conversion ratio.

To counteract the *C. jejuni* infection, Arsi et al. (2014) [162] investigated the use of thymol, carvacrol, or a combination thereof as a dietary supplement. In order to perform this, four separate studies were conducted with day-old broiler chicks (n = 10 chicks/dose) given thymol, carvacrol, or mixtures of these compounds in feed. On day three, the birds were orally challenged with *C. jejuni*, and on day ten, cecal samples were taken in order to count the number of *Campylobacter*. The study found that *Campylobacter* levels decreased for thymol treatments at 0.25% (trial 1), 1% carvacrol or 2% thymol (trial 2), or a combination of both thymol and carvacrol at 0.5% (trial 3). These findings back up the addition of these substances to feed in an effort to lessen hens’ colonization of *Campylobacter.* Thymol at a 0.25% concentration decreased *Campylobacter* by 0.6 log CFU/mL cecal contents. With 2% thymol, a 2 log CFU decrease was seen. Furthermore, a single attempt employing 0.5% thymol and carvacrol resulted in 2 log CFU/mL cecal contents.

*Campylobacter* species can also infect mammals. Specifically, dogs and cats may show asymptomatic or develop clinical signs related to the gastrointestinal tract, but in any case, they pose a risk of infection to their owners. Although there are no studies comparing the effectiveness of EOs to pet strains of *Campylobacter* in the literature, it is plausible that isolates from dogs and cats may be susceptible to EOs that have been shown to be effective against strains of *Campylobacter* that originate from other sources.

### 4.2. Gram-Positive Bacteria

#### 4.2.1. *Staphylococcus* Species

*Staphylococcus* species are well-known to cause opportunistic infections in humans, but they also pose a serious threat in veterinary medicine. These pathogens infect cold-blooded animals, birds, and mammals in many anatomical areas.

Both coagulase-positive and coagulase-negative staphylococci, which are implicated in infections of varying degrees of severity, belong to this genus. Among coagulase-positive species, the most well-known is *S. aureus*. Staphylococci are the most common saprophytes of human and animal skin. Specific sites of colonization are, in cattle, the skin of the teats and udder. They are found in the nostrils and tonsils of cattle and sheep, on the skin of the head and neck, and in the external auditory meatus of pigs. In the presence of predisposing factors, they pass through the skin, causing mild infections. They are also responsible for specific diseases: mastitis, pyaemia in lambs, botryomycosis, endometritis, exudative epidermitis, pyoderma, and otitis. Numerous infections in humans, including moderate superficial skin infections, osteomyelitis, native valve endocarditis, heart valve implant-associated infections, severe sepsis, and bacteremia, are caused by staphylococci. *S. aureus* diseases are typically linked to mastitis in animals raised for dairy products. *S. pneumoniae* commonly infects companion animals. *S. pseudointermedius* is a component of the mucosal, cutaneous, and upper respiratory tract microflora, but it has the potential to develop into an invasive pathogen that may identify several illnesses. Still, some species, such as *S. chromogenes*, *S. xylosus*, and *S. hyicus*, have the ability to infect pets. Staphylococcal infections in agricultural animals may pose a serious financial risk.

Numerous investigations have confirmed EO’s anti-staphylococcal activity. Good anti-staphylococcal action was shown by *Satureja montana* EO [150]. Prior observations of *S. montana* EO’s antibacterial efficacy against a variety of Gram-positive and Gram-negative bacteria established a connection with the compound’s main ingredients, including carvacrol. Vitanza et al. (2019) [163] pointed out that *S. aureus* exposed to the EO of *Satureja montana* underwent cell wall disruption.

Ebani et al. (2020) [164] discovered that whereas EOs from *C. zeylanicum* and *C. myrrha* showed no action against *S. xylosus*, they exhibited modest activity against many canine isolates of *S. aureus*, *S. pseudointermedius*, *S. chromogenes*, and *S. hyicus*. In a similar vein, *S. aureus* was found to be somewhat sensitive to myrrh oil by Adam and Selim (2013) [165]. Furthermore, Mahboubi and Kazempour (2016) [166] found that *Commiphora myrrha* has relevant efficacy against *S. aureus*.

Studies have demonstrated that *S. aureus* is susceptible to the EOs of *Cymbopogon citratus* and *Melissa officinalis*. *Aloysia citrodora* EO also reduced the development of *S. aureus* reference strains by compromising the plasmic membrane’s structural integrity and causing a loss of cytoplasmic contents, which results in cellular death.

Most generally, *S. aureus* is regarded as the most significant microbe connected to mastitis. Because of their propensity for chronicity and recurrence, intramammary infections brought on by this pathogen are extremely difficult to treat [167]. The substantial economic effect of *S. aureus*-caused bovine mastitis is a major problem in veterinary practice. In an in vivo investigation, Abboud et al. (2015) [168] found that the EOs of *Thymus vulgaris* and *Lavandula angustifolia* exhibited potent antibacterial activity against strains of *Streptococcus* and *Staphylococcus*. After four consecutive days of therapy, an intramammary application of these oils, or a combination of them, resulted in a significant reduction in the number of bacteria in the various milk samples. In the same study, the external use of *Thymus* essential oil in Vaseline produced greater antibacterial action, leading to a 100% recovery rate. In a different trial conducted by Cho et al. in 2015 [169], single and double dosages of *Origanum* EO ointment (0.9 mL of EO in a 10 mL tube) were infused into the affected quarters twice a day for three days. The circumstances of the udders improved. Somatic cell counts considerably decreased in the groups as compared to pre-treatment levels. Additionally, on the fourth day following the therapy, no *S. aureus* or *E. coli* were found in the milk for three days. Comparable effectiveness was noted for white blood cell counts, which sharply dropped in comparison to pre-treatment levels [169].

#### 4.2.2. *Streptococcus* Species

Streptococci are widespread in nature as commensals and parasites of the oral cavity, intestinal, and genito-urinary tracts of humans and animals. Pathogenic species are responsible for specific diseases. For example, *S. agalactiae* causes mastitis in ruminants, while *S. equi* is responsible for adenitis in horses. The majority of the information found in the literature relates to the effectiveness of EOs against strains of human-originating *Streptococcus* spp. While some streptococcal species—like *S. pyogenes*—can infect humans as well as animals, other streptococcal species primarily cause infections and illnesses in a variety of companion and farm animals.

Testing cinnamon oil extracts from *Cinnamomum verum* and *Neolitsea cassia* revealed antimicrobial efficacy against a number of diseases, including certain *Streptococcus* species [170,171,172].

Cinnamaldehyde, the primary constituent of *Cinnamomum* essential oil, has been linked to the antistreptococcal action of the plant against other bacteria.

*S. iniae* may affect a wide variety of freshwater, estuarine, and marine fish species. Among the most vulnerable species are *Oncorhynchus mykiss*, *Seriola quinqueradiata*, *Lates calcarifer*, and *Oreochromis niloticus* [173,174,175,176,177]. A fish infection that is spreading over the world and is devastating many freshwater and saltwater species financially is sustained by *Streptococcus agalactiae* [178]. The bacterium has been found in a variety of fish, including rainbow trout, seabream, tilapia, yellowtail, croaker (*Micropogonius undulatus*), killfish (*Menhaden* spp.), and silver pomfret (*Pampus argenteus*).

Serious zoonotic pathogens include some streptococci species. For example, *S. iniae* can cause bacteremia, cellulitis, meningitis, and osteomyelitis [179], *S. agalactiae* can cause neonatal meningitis, sepsis, and pneumonia [180,181]. Both *S. dysgalactiae* subsp. *equisimilus* and *S. dysgalactiae* subsp. *dysgalactiae* can cause bacteremia, lower limb cellulitis, and meningitis in infants.

In vitro, the ethanolic extract of *Thymus daenensis* was less effective in controlling *S. iniae* than its EO [182]. With the lowest minimum inhibitory concentrations (MICs) of 39 μg/mL and 31.2 μg/mL, respectively, the EO of Bakhtiari savory (*Satureja bachtiarica*) showed strong action against both *S. iniae* and *S. agalactiae* strains [166]. *Aloe vera* EO showed no inhibitory action against *S. iniae* strains that were first isolated from sick rainbow trout [183], but its ethanolic extract had an anti-*S. iniae* impact that was similar to that of erythromycin.

Because of its bacteriostatic action, *Lavandula angustifolia* EO has been demonstrated to be a potential inhibitor against isolates of *S. iniae* and *S. parauberis* [184]. With a MIC of 0.063% and a MIC:MBC ratio of 1:8, one strain of *S. parauberis* was the most sensitive. This suggests that, while it cannot survive at extremely low concentrations of lavender EO, there may be a chance for bacterial survival at high concentrations.

The fish pathogen *S. iniae* has been shown to be effectively inhibited by *Cinnamomum verum* both in vitro and in vivo. Five days before *S. iniae* infection, tilapia given fish diets supplemented with 0.4% (*w*/*w*) of cinnamon oil and 0.1% (*w*/*w*) of oxytetracycline showed no signs of death during an in vivo study [172].

In an in vivo investigation, the modifying impact of clove *Syzygium aromaticum* EO on the antioxidant and immunological state of Nile tilapia after *S. iniae* infection was assessed. Fish immunity was boosted by clove EO against bacterial assault. Furthermore, it was shown that this oil’s antibacterial activity was partially mediated via hepatic hepcidin expression [185].

De Aguiar et al. (2019) [186] demonstrated the in vitro efficacy of oregano and *Thymus* spp. EOs against *S. suis* isolates.

#### 4.2.3. *Mycobacterium* Species

Although some research has been conducted to confirm the efficacy of EOs against *Mycobacterium tuberculosis* and *M. bovis*, the species that cause tuberculosis in both humans and animals [187], the use of alternative treatments for tuberculosis is discouraged given the seriousness of this infectious disease.

Conversely, EOs may serve as substitute natural treatments for diseases brought on by non-tuberculous mycobacteria (NTM).

These are opportunistic bacteria that are found in soil and water and are widely distributed in the environment. They are often spread via skin infection, eating or drinking infected food or water, and inhaling aerosol particles. A number of NTM strains generate biofilm, which increases their resistance to antimicrobial drugs, disinfectants, and environmental stressors [188]. Human forms caused by NTM include disseminated forms, pulmonary mycobacteriosis, skin infections, and lymphadenitis. These mycobacteria produce identical infections in a number of animal species, which makes them significant in veterinary medicine as well. Dogs and cats’ cutaneous forms are often linked to NTM species.

Conventional medication therapy is employed, although it’s not always effective. Because of this, using EOs topically may be a viable substitute; however, more research is required to determine the EOs’ additional in vivo functions in addition to their lack of toxicity. Even though *M. avium* is linked to TB in birds, it is still considered an NTM agent.

In vitro experiments were conducted to show the antibacterial and antibiofilm properties of *Juniperus communis*, *Helichrysum italicum*, and *L. hybrida* EOs against NTM, namely *M. avium*, *M. intracellulare*, and *M. gordonae* [188,189].

EO from *Zingiber officinale*, which is distinguished by a high concentration of monoterpenes and sesquiterpenes, has shown efficacy against a few NTM species, namely *M. abscessus* subsp. *massiliense* and *M. chelonae* [187].

Research has been conducted to confirm the in vitro efficacy of a few EOs against *M. avium* subsp. *paratuberculosis*, an NTM pathogen that causes paratuberculosis, a serious illness that causes significant economic losses in ruminants. *Myrtus communis* EO had poor efficacy [91], whereas oils of cinnamon and oregano showed strong action [190,191].

The infection known as *M. avium* subsp. *paratuberculosis* is found in the udders and the gastrointestinal tract. While it is unlikely that EOs might be utilized as a medication to treat ruminants afflicted with paratuberculosis, natural products, which have shown efficacy in vitro, may be used to lessen the pathogen’s ambient farm contamination.

The botanical species effective against the bacterial species considered in this manuscript are summarized in Table 3.

## 5. Limitations and Future Perspective

Although there appears to be a strong link between medicinal plant extracts and antibacterial activity, the majority of current research is based on in vitro investigations; therefore, it is unclear how applicable this information would be in a clinical context. In vivo, compounds with antibacterial activity in vitro could not have much of an impact. The primary impediment to the application of medicinal plants as antimicrobials is the absence of treatment standardization. This is one of the reasons why the effectiveness of medicinal herbs is not widely believed. The link between the structure and activity of bioactive chemicals, as well as their methods of action, has largely remained unknown until recently. Standardizing treatment protocols will result from learning more about the pharmacology of substances obtained from medicinal plants [192]. There are few clinical studies assessing the efficacy of medicinal plant components for treating infectious disorders and identifying their side effects [193,194]. Standardized testing is necessary to evaluate the antimicrobial activity of pure bioactive chemicals; however, more sensitive bioassay approaches are required to test plant extracts or EOs [195]. The repeatability of plant extract composition is another significant constraint. It is well recognized that, depending on the suppliers, an extract might have various qualities. To create quality control methods, bioactive substances must be accurately characterized and authenticated [196,197]. In order to demonstrate a reliable association between in vitro effectiveness results and the therapeutic relevance of these drugs, it will be necessary to conduct in vivo research on animal models in the future [198]. For both skin and mucous membranes, topical application is currently the most promising approach [199,200]. The ability of EOs to increase the penetration of antiseptics might be used to both restore antibacterial action against resistant species and prevent surgical and medical device-related infections [201,202]. Although clinical testing is required, there is some potential for inhalation applications [203].

It is necessary to learn more about the pharmacokinetics, pharmacodynamics, and possible toxicity of EOs taken orally because there is currently little evidence about the safety of this method of administration. The acceptability of EOs or EO component supplements in animal feed has not been thoroughly studied [204]. This is typically seen when strong-tasting and odorous ingredients are used, including allicin and allyl isothiocyanate [205]. Inevitably, feed supplementation techniques (such as encapsulation) could be used to prevent these side effects.

With the application of EOs and substances like carvacrol and cinnamonaldehyde that have more agreeable sensory qualities. Due to their greater number of taste receptors, some animals, like pigs, are actually more sensitive to changes in flavor characteristics [206]. However, certain methods that make it easier to add EOs to animal feed may raise manufacturing costs, making industrial applications less feasible. Actually, for economic reasons primarily, certain manufacturers exhibit opposition to the use of EOC and other technologies. The widespread usage of antibiotics is a result of their affordability and convenience of use [204].

Compound structural alterations to enhance pharmacokinetics and pharmacodynamics, as well as evaluations of the structure-activity connection, should be included in future research. In order to elucidate the mechanism behind these compounds’ antimicrobial activity and identify other pathways that may be targeted, further research should be performed on the synergistic interactions seen in medicinal plant extracts and between substances and antibiotics. Antimicrobial drugs and medicinal plant extracts can interact in ways that are beneficial (synergism) or detrimental (antagonistic) [207]. To qualify these chemicals as biomedical agents, more study is needed, particularly on their toxicity and in vivo investigations.

## 6. Discussion

In the 1940s, researchers found that some antibiotics, such as oxytetracycline, promoted animal development when they were introduced for the treatment of bacterial infections in both people and animals [208]. Following research showing that the use of both oxytetracycline and chlortetracycline enhanced feed utilization, decreased mortality and morbidity from clinical and subclinical infections in animals, and increased growth rate, the Food and Drug Administration approved their use as additives in animal feed in 1951–1953 [208,209].

Sub-therapeutic tetracycline and other antibiotic administrations to farm animals have contributed to the development of antibiotic resistance in humans, according to the Swann report from 1969, which raised concerns about the use of antibiotics as growth promoters [208]. Because it is frequently linked to the animals involved in acquiring resistant enteric flora, which in turn adds to the human reservoir of antimicrobial-resistant coliforms and Salmonella, the practice has been under constant scrutiny since then because of the possible risks involved [209]. Since antibiotic resistance was discovered to be transferable more than 30 years ago, the use as growth promoters of those antibiotics for which cross-resistance has been demonstrated has been prohibited [210].

Antibiotic resistance in animal microorganisms and the establishment of antibiotic resistance in human pathogens are believed to be influenced by the use of antibiotics in veterinary medicine. There are several instances when bacteria or their genes are transferred from animals to humans via the food chain [211]. Resistance gene transfer across bacterial populations occurs via both straightforward and intricate pathways within the ecology of bacteria. Furthermore, there is evidence that resistance genes from humans have returned to the animal population, supporting the idea that resistant microorganisms may spread from animals to people. The possibility of these genes being amplified in animal populations makes this a serious problem. New antimicrobial agents need to be developed.

The pharmaceutical industry is said to generate two or three novel antibiotics from microbes annually on average [212]. As the number of novel antimicrobial medications in the research and development (R&D) pipeline has started to decline over the past 20 years, the pharmaceutical industry has grown more accepting of the potential applications of plant-derived drugs [212]. In addition, the general public is now very interested in complementary therapies like “medicinal plants” due to growing awareness of the abuse and overuse of antibiotics. An estimated 20–80% of people use and believe that botanical products are safe and beneficial in many different places around the globe. Therefore, “traditional knowledge,” or the variety of information gathered by indigenous peoples on the plant and animal items they have utilized to cure illnesses and preserve health, is of enormous interest to people all over the world [212].

In addition to boosting the biosafety of breeding stock, the advantageous impacts of Eos may also be used to substitute performance-enhancing additives, as shown by the zootechnical indices. For example, when EO extracted from *Poliomintha longiflora* (400 mg L^−1^) was added to chicken feed, Hernández-Coronado et al. (2019) [213] demonstrated improved sensory evaluation of chicken meat. When broiler feed was supplemented with 200 mg kg^−1^ of nanoencapsulated cumin EO with chitosan, it produced better body weight gain (BWG) and feed conversion ratio than 650 mg kg^−1^ of flavomycin [214]. *Eucalyptus globulus* was able to lower the *E. coli* population and encourage the formation of beneficial bacteria. Additionally, it boosted the digestibility of organic matter, which can enhance nutritional absorption, lower blood cholesterol, and boost superoxide dismutase activity. While cholesterol reduction improved the meat’s nutritional profile and increased BWG and feed conversion, superoxide dismutase’s inhibition of free radicals increased antioxidant activity [215].

A combination of 300 mg/day of Eos from *Thymus kotschyanus*, *Lavandula angustifolia*, *Salvia officinalis*, and *Capparis spinosa* was used in another experiment on calves. The results demonstrated an improvement in animal performance (59.1 kg final body weight for the control group and 62.3 kg final body weight for the EO group) because of the antioxidant and bactericidal properties of the mixture. The information provided suggests that it is quite likely that EOs will be used in the creation of biotechnological substitutes for traditional therapies. In addition to having possible antibacterial properties, the use of EOs in chicken diets is of interest as it seems to enhance egg quality [216].

Moreover, EOs are incredibly adaptable substances that have the ability to function as immunomodulators, detoxifiers, and performance boosters. Research should be done to confirm the adverse effects of administering EOs in vivo. Cytotoxicity and therapeutic index assessments must come first. Dušan et al. (2006) [217] proved the toxicology of several EOs towards intestinal cell viability. Dosage dependence was seen in the antimicrobial activity of certain plant extracts against enteroinvasive *E. coli*. In vitro cultures of intestinal-like cells revealed comparatively substantial cytotoxicity to EO concentrations that could totally block bacterial growth (0.05%). Reduced EO concentrations (0.01%) only partially inhibited microbes and had a negligible negative impact on Caco-2 cells. The essential oils of clove and cinnamon, as well as their main component, eugenol, had almost no cytotoxic impact at lower concentrations, according to an assessment of cell deathbased on morphological and viability staining followed by fluorescence microscopy. Despite a small increase in the incidence of apoptotic cell death, oregano essential oil and its constituent carvacrol demonstrated broad antibacterial action even at lower doses. Thyme oil showed a relatively significant level of cytotoxicity, increasing the incidence of both necrotic and apoptotic cell death. On the other hand, the constituent thymol has shown neither cytotoxic impact nor significantly diminished capacity to impede the pathogen’s apparent development at the employed levels [217].

## 7. Conclusions

There is a wealth of scientific data supporting traditional knowledge on the antibacterial properties of plant extracts. A simple search using the PubMed database reveals hundreds of publications in the scientific and medical literature describing the antibacterial properties of various plant species and their chemical components. Furthermore, 6345 plant species show experimental antibacterial activity, according to a search of the Napralert database, which is a database of natural products maintained by the University of Illinois at Chicago and contains 58,725 plant species. It is very likely that plant-based antimicrobial agents can replace traditional antibiotics. In this respect, EOs can also be used to reduce drug resistance. For example, some attempts have been made to improve or restore antibacterial efficacy against multidrug-resistant germs. Antimicrobial MICs can be reduced by adding essential oils to antibiotics; aminoglycosides, including amikacin, have shown the greatest reduction in MICs [218]. Their use has also been shown to restore drug sensitivity in strains that overexpress efflux pumps [143]. The way Eos modify drug resistance is particularly evident when it comes to drugs such as fluoroquinolones, β-lactams, and chloramphenicol. This manuscript, therefore, is particularly interesting as a catalyst for a new green pharmacology that triggers virtuous practices useful for what is often referred to as “One-health”.

## Figures and Tables

**Figure 1 antibiotics-13-00163-f001:**
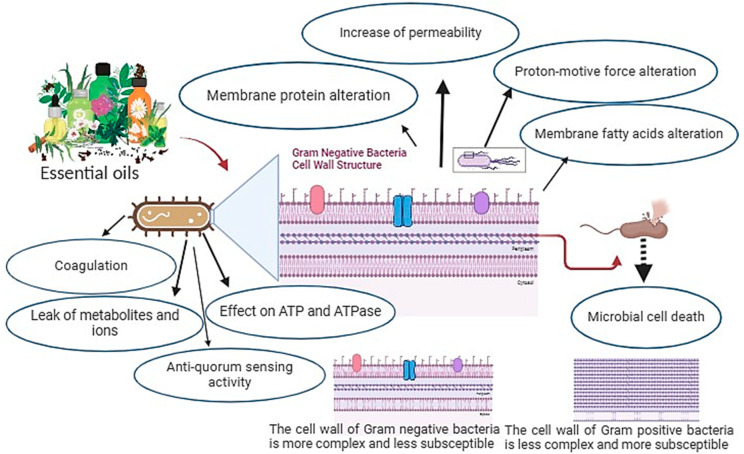
Mechanisms of EOs action that lead to microbial cell death.

**Table 1 antibiotics-13-00163-t001:** Essential oils and essential oil components with antimicrobial activity.

EO or EO Component	Bacteria	Mechanism	Reference
(−)-Borneol	*Escherichia coli* *Staphylococcus aureus* *Salmonella typhimurium*	Cell membrane breakdown	[54]
(+)-Borneol	*E. coli* *S. aureus*	Cell membrane breakdown	[54]
Citral	*E. coli* *S. aureus* *S. typhimurium*	Cell membrane breakdown	[54]
Citronellal	*E. coli* *S. aureus* *S. typhimurium*	Cell membrane breakdown	[54]
Hibicuslide C	*P. aeruginosa*	DNA fragmentation	[55]
Quercetin	*P. aeruginosa*	Inhibition of FabZ enzyme	[56]
Thymol	*E. coli* *S. aureus* *S. typhimurium*	Cell membrane breakdown	[54]
*Cuminum cyminum*	*E. coli*	Cell membrane breakdown	[57]
*Mentha piperita*	*Streptococcus mutans*	Cell membrane breakdown and cell leakage	[58]
*Origanum vulgare*	*S. aureus*	Cell membrane breakdown	[57]
*Pimenta dioica*	*P. aeruginosa*	Cell membrane breakdown and leakage of K+ from cytosol	[59]
*Solidago canadens*	*Pseudomonas fluorescens* *S. mutans*	Cell wall breakdown	[60]

**Table 2 antibiotics-13-00163-t002:** Important bacteria causing microbial infections and classes of antibiotics to which resistance has been recorded.

Bacterial Agent of Infection	Common Resistance
*Escherichia* spp.	Cephalosporins, fluoroquinolones, aminoglycosides [64]
*Salmonella* spp.	Tetracyclines, Sulfonamides, Streptomycin, Kanamycin, Chloramphenicol, β-lactams, Amoxicillin/clavulanic acid, Nalidixic acid, Ceftriaxone [65]
*Klebsiella* spp.	Cephalosporins, fluoroquinolones, aminoglycosides, carbapenems [66]
*Pseudomonas* spp.	Piperacillin/tazobactam, ceftazidime, ciprofloxacin, aminoglycosides [67]
*Campylobascter* spp.	Quinolones, Macrolides, Lincosamides, Chloramphenicol, Aminoglycosides, Tetracycline, β-lactams, Cotrimoxazole, Tylosin [68,69]
*Staphylococcus* spp.	β-lactams, fluoroquinolones, macrolides, aminoglycosides [70]
*Streptococcus* spp.	β-lactams, macrolides, tetracyclines, co-trimoxazole [71,72]
*Mycobacterium*	Rifampin, isoniazid, and three of the following: aminoglycosides, polypeptides, fluoroquinolones, thioamides, cycloserine, or para-aminosalicylic acid [73,74]

**Table 3 antibiotics-13-00163-t003:** Essential oils used for the control of the main bacterial etiological agents causing diseases in zootechnical species.

Bacterial Species	Botanical Species	References
*Escherichia coli*	*Aloysia triphylla*, *Backhousia citriodora*, *Cananga odorata*, *Neolitsea cassia*, *Cinnamomum verum*, *Citrus limon*, *Cymbopogon citratus*, *Cymbopogon nardus*, *Cymbopogon martini*, *Cupressus sempervirens*, *Daucus carota*, *Eucalyptus globulus*, *Foeniculum vulgare*, *Juniperus communis*, *Litsea cubeba*, *Mentha piperita*, *Melissa officinalis*, *Ocimum basilicum*, *Origanum majorana*, *Origanum vulgare*, *Pimenta racemosa*, *Pelargonium graveolens*, *Pimpinella anisum*, *Pinus sylvestris*, *Syzygium aromaticum*, *Thymus vulgaris*, *Thymus zygis*, *Zingiber officinale*	[81,83,84,85,87,88,89,91,96]
*Salmonella* spp.	*Aloysia triphylla*, *Aristolochia fontanesii*, *Artemisia dracunculus*, *Cinnamomum verum*, *Calamintha nepeta*, *Cinnamomum verum*, *Citrus sinensis*, *Cymbopogon citratus*, *Syzygium aromaticum*, *Eucalyptus globulus*, *Eucalyptus exserta*, *Litsea cubeba*, *Mentha piperita*, *Origanum virens*, *Pimenta pseudocaryophyllus*, *Satureja montana*, *Syzygium aromaticum*, *Spondias pinnata*, *Taxodium distichum*, *Thymus mastichina*, *Thymus vulgaris*	[101,104,105,106,107,108]
*Klebsiella* spp.	*Alstonia scholaris*, *Arbutus unedo*, *Calicotome villosa*, *Cistus* spp., *Cytinus* spp., *Cytinus hypocistis*, *Crinum angustum*, *Melaleuca alternifolia*, *Myrtus communis*, *Pistacia lentiscus*, *Pistacia terebinthus*, *Rapa catozza*, *Satureja kitaibelii*, *Teucrium chamaedrys*, *Rhus coriaria*, *Thymus broussonetii*, *Thymus maroccanus*, *Thymus pulegioides*, *Thymus vulgaris*, *Tinospora cordifolia*	[120,121,122,123,124,125,126,127,128,129,130,131,132,133,134,135,137]
*Pseudomonas* spp.	*Abies alba*, *Carum carvi*, *Cinnamomum aromaticum*, *Cinnamomum verum*, *Citrus aurantium*, *Coriandrum sativum*, *Cymbopogon nardus*, *Foeniculum vulgare*, *Illicium verum*, *Lavandula angustifolia*, *Lavandula latifolia*, *Litsea cubeba*, *Melaleuca alternifolia*, *Mentha piperita*, *Mentha spicata*, *Myroxylon balsamum*, *Ocimum basilicum*, *Origanum vulgare*, *Pimpinella anisum*, *Pinus cembra*, *Pinus sylvestris*, *Rosmarinus officinalis*, *Salvia sclarea*, *Satureja hortensis*, *Syzygium aromaticum*, *Thymus serpyllum*, *Thymus vulgaris*	[148,149,150,151]
*Campylobacter* spp.	*Apium graveolens*, *Artemisia vulgaris*, *Backhousia citriodora*, *Calendula arvensis*, *Citrus aurantium*, *Daucus carota*, *Jasminum officinale*, *Melaleuca alternifolia*, *Nardostachys jatamansi*, *Origanum syriacum*, *Pogostemon cablin*	[157,158,159]
*Staphylococcus* spp.	*Achlys triphylla*, *Cinnamomum verum*, *Cymbopogon citratus*, *Lavandula angustifolia*, *Melissa officinalis*, *Origanum* spp., *Satureja montana*	[150,164,165,168,169]
*Streptococcus* spp.	*Neolitsea cassia*, *Cinnamomum verum*, *Lavandula angustifolia*, *Origanum* spp., *Satureja bachtiarica*, *Syzygium aromaticum*, *Thymus daenensis*	[170,171,172,182,183,185,186]
*Mycobacterium* spp.	*Helichrysum italicum*, *Juniperus communis*, *Lavandula latifolia*, *Myrtus communis*, *Zingiber officinale*	[165,188,189,190,191]

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
