# Peer review of "Use of Essential Oils to Counteract the Phenomena of Antimicrobial Resistance in Livestock Species"

_antibiotics, 2024, doi:10.3390/antibiotics13020163_

Round 1
Reviewer 1 Report
Comments and Suggestions for Authors
Title: Use of essential oils to counteract the phenomena of antimicrobial resistance in livestock species
The review “Use of essential oils to counteract the phenomena of antimicrobial resistance in livestock species” aims to shed light on the results achieved by the scientific community regarding the use of essential oils for the treatment of the main bacterial genera of veterinary interest in livestock. This manuscript, therefore, is particularly interesting as a catalyst for a new green pharmacology that triggers virtuous practices useful for what is often referred to as One health. It is well written article with some interesting findings; however, there are some corrections before its acceptance for publication:
Line 26: The pharmacological properties of the essential oils should be discussed in the abstract portion.
Line 31: Authors should also describe the methods part in the abstract i.e., how many research articles were consulted for this review?
Line 59: Describe the change in eating trends of humans, as they are demanding organic animal products i.e., antibiotic-free meat or milk.
Line 70: Along with essential oils, discuss the importance of supplementation of probiotics and/or prebiotics, the use of bacteriophages as alternatives to minimize the use of antibiotics in animal husbandry.
Line 162: It is an important and interesting section for the readers, therefore, compile the information in a Table form describing different essential oils with their active ingredients and mechanism of action with the reference.
Line 224: Before discussing each bacterium separately, authors should draw a table describing the antibiotic and their resistance against the bacteria, it would be very interesting and appealing for the readers based on the title of the review.
Line 547: Authors should also use the common names of all essential oils mentioned in the review article that would be easy for the readers to understand, at other places in the manuscript as well.
Line 778: Gram-negative bacteria exhibit higher resistance to the effects of essential oils and their components due to the distinct composition of their cell wall as compared with the Gram-positive bacteria. Therefore, I invite the authors to draw a figure describing the mechanism of action and/or how essential oils kill the bacteria. I recommend the authors consult following paper for it:
· Stojiljkovic, J.; Trajchev, M.; Nakov, D. et al. Antibacterial activities of Rosemary essential oils and their components against pathogenic bacteria. Adv Cytol Pathol. 2018, 3(4):93-96. DOI: 10.15406/acp.2018.03.00060.
The figure may be included in discussion segment that will make it more meaningful.
I invite the authors to read the following reference in detail to strengthen the discussion part of the manuscript:
· Evangelista, A. G., Corrêa, J. A. F., Pinto, A. C. S. M., & Luciano, F. B. (2022). The impact of essential oils on antibiotic use in animal production regarding antimicrobial resistance–a review. Critical Reviews in Food Science and Nutrition, 62(19), 5267-5283.
Line 873: After conclusions, there should be a separate section at the end of the manuscript describing the futuristic visions Use of essential oils to counteract the phenomena of antimicrobial resistance in livestock species.
English grammar and sentence structure should be revised and corrected throughout the manuscript. In many places, the sentences are too long those are difficult to understand.
Comments on the Quality of English Language
English grammar and sentence structure should be revised and corrected throughout the manuscript. In many places, the sentences are too long those are difficult to understand.
Author Response
Title: Use of essential oils to counteract the phenomena of antimicrobial resistance in livestock species
The review “Use of essential oils to counteract the phenomena of antimicrobial resistance in livestock species” aims to shed light on the results achieved by the scientific community regarding the use of essential oils for the treatment of the main bacterial genera of veterinary interest in livestock. This manuscript, therefore, is particularly interesting as a catalyst for a new green pharmacology that triggers virtuous practices useful for what is often referred to as One health. It is well written article with some interesting findings; however, there are some corrections before its acceptance for publication
R: Thank you for your attention to our manuscript and for your important revision work that helped us to improve the overall quality of the document. We have made the requested changes which you can find highlighted in the text.
Line 26: The pharmacological properties of the essential oils should be discussed in the abstract portion.
R: Thank you for your suggestion. We have included a sentence regarding the pharmacological properties of essential oils in the abstract.
Line 31: Authors should also describe the methods part in the abstract i.e., how many research articles were consulted for this review?
R: Thanks for this tip. We have included a paragraph in which the methods used to produce the manuscript were discussed.
Line 59: Describe the change in eating trends of humans, as they are demanding organic animal products i.e., antibiotic-free meat or milk.
R: Thank you for this important advice that helps us improve the quality of the manuscript. We have discussed on this topic at line 79.
Line 70: Along with essential oils, discuss the importance of supplementation of probiotics and/or prebiotics, the use of bacteriophages as alternatives to minimize the use of antibiotics in animal husbandry.
R: Thank you for this valuable advice which helps us improve the quality of the manuscript, we have included information on these elements at line 83.
Line 162: It is an important and interesting section for the readers, therefore, compile the information in a Table form describing different essential oils with their active ingredients and mechanism of action with the reference.
R: Thanks for this valuable advice. A table reporting these information has been added.
Line 224: Before discussing each bacterium separately, authors should draw a table describing the antibiotic and their resistance against the bacteria, it would be very interesting and appealing for the readers based on the title of the review.
R: Thanks for this suggestion. A table reporting these information has been added.
Line 547: Authors should also use the common names of all essential oils mentioned in the review article that would be easy for the readers to understand, at other places in the manuscript as well.
R: We preferred to keep only the scientific names (which have been checked and standardized throughout the text) and eliminated all common names. We ask for your approval for this choice which we believe is the most suitable to confer greater scientific value to the manuscript.
Line 778: Gram-negative bacteria exhibit higher resistance to the effects of essential oils and their components due to the distinct composition of their cell wall as compared with the Gram-positive bacteria. Therefore, I invite the authors to draw a figure describing the mechanism of action and/or how essential oils kill the bacteria. I recommend the authors consult following paper for it: Stojiljkovic, J.; Trajchev, M.; Nakov, D. et al. Antibacterial activities of Rosemary essential oils and their components against pathogenic bacteria. Adv Cytol Pathol. 2018, 3(4):93-96. DOI: 10.15406/acp.2018.03.00060.
R: The existing image was modified. In its corner, the structural differences of parasites between Gram positive and Gram negative bacteria were mentioned and represented. In particular, as you can now see, the Gram negative wall was drawn which is more complex and the Gram positive wall which is simpler.
The figure may be included in discussion segment that will make it more meaningful.
R: With your agreement we would prefer to keep the figure in the position it was originally placed. We believe that, since it is a representation of the action exerted by essential oils on the bacterial cell, it should be kept in the "Antimicrobial mechanism of action" paragraph and not in the discussions.
I invite the authors to read the following reference in detail to strengthen the discussion part of the manuscript: Evangelista, A. G., Corrêa, J. A. F., Pinto, A. C. S. M., & Luciano, F. B. (2022). The impact of essential oils on antibiotic use in animal production regarding antimicrobial resistance–a review. Critical Reviews in Food Science and Nutrition, 62(19), 5267-5283.
R: Thank you for this important advice. Several elements of this manuscript gave us pause. The reflections arose in the new paragraph: "Limitations and future perspective". In this paragraph, the manuscript you recommended for consultation has been cited.
Line 873: After conclusions, there should be a separate section at the end of the manuscript describing the futuristic visions Use of essential oils to counteract the phenomena of antimicrobial resistance in livestock species.
R: Thanks for this important advice. We have developed these concepts in the new paragraph entitled “Limitations and future perspective”.
English grammar and sentence structure should be revised and corrected throughout the manuscript. In many places, the sentences are too long those are difficult to understand.
R: The English language was improved in accordance.

Reviewer 2 Report
Comments and Suggestions for Authors
Reviewer’s comments:
The authors presented the antimicrobial resistance of essential oils. I found that the obtained results are interesting. However, there are some deficiencies in the manuscript. Despite the descriptive writing of the obtained results, it is better to provide those data with graphical illustrations or comprehensive tables.
· Binomials of the plants must be validated through the authentic website “the World Flora Online” (WFO)
· Authors must include the part from which the essential oil is extracted.
· Author citations for all the plant species should be given on their first mention.
· Mechanism of microbial action is given for gram-negative bacteria alone, adding an illustration for gram-positive bacteria too is required.
· Graphical representations, illustrations, and tables for the obtained data can be more meaningful to the readers.
· The Discussion part is poorly written. So, the authors should enrich this section with recent relevant literature.
· In the table authors have mentioned family names for some species alone, try to include for all the species.
· Line 65 Justify the sentence.
· Lines 119, 123-126, 128-130, 214, 297-298, 306-307, 311-313, 325, 607, 618-619, 631, 655-657, 662, 712, 734 Rephrase all these sentences without grammatical errors.
· 131 family
· Lines 230 and 235 specific epithets must be in lower case.
· Line 246 what does contaminating eggs represent the sentence?
· Line 289 how E. coli could show inhibition?
· Line 316 check the binomial name for citrus lemon.
· Line 317 check the spelling mistake in binomial and author citation Zingiber officinalis Rosc.
· Line 319 scientific name must be in italics
· Line 337 check the spelling of Salmonella and italics.
· Line 338 Check the binomial name ‘S. abortive ovis’
· Line 340 full stop missing after genus abbreviation ‘S pullorum’
· Lines 353, 354 check the scientific names and italics it.
· Line 359 Uppercase in between sentence must be avoided
· Combine the sentence 387 and 388 with relevant paragrapgh
· What is that term RCMB in lines 417, 425
· Check the author citation for the binomial line 431
· Line 448 Write the binomial with author citation and check its current name.
· Line 489 italics for the scientific name is required.
· 521, 527 full stop is missing between abbreviated genus and species name.
· Line 528 and 529 mention the part from which essential oil is extracted and the sentence must be rephrased.
· Line 544 Et al. isn’t plural already the word itself is self-explanatory.
· Line 555 author shouldn’t include the year of publication for the references cited in the running text according to author guidelines. Stick to the pattern of the journal.
· Line 600 check the spelling for the bacterial strain
· Line 618, 619 what does the authors mean by the usage of word ‘just’?
· Line 649 species name must not start with uppercase
· Line 662 Usage of ATCC is irrelevant, rephrase the sentence
· Line 677 include the concentration
· Lines 755, 759 M. avium avium species name is repeated
· Line 767 Denote the term sub species
Reference style in running text must be checked with author guidelines and change it in the required places.
Author Response
The authors presented the antimicrobial resistance of essential oils. I found that the obtained results are interesting. However, there are some deficiencies in the manuscript. Despite the descriptive writing of the obtained results, it is better to provide those data with graphical illustrations or comprehensive tables.
Dear reviewer, thank you for taking the time to review us and thank you also for the comments given. We have corrected as indicated, below you will find all the answers to your sentences.
R: Thanks for the valuable advice. More tables have been inserted in the text to make consultation quicker.
- Binomials of the plants must be validated through the authentic website “the World Flora Online” (WFO)
R: checked and standardized throughout the text
- Authors must include the part from which the essential oil is extracted.
R: We followed the advice and we added, where possible, the parts from which the EO was extracted. Unfortunately, this was not always possible to do; in some cases, the part of the botanical species used was not even specified in the original text.
- Author citations for all the plant species should be given on their first mention.
R: The bibliography has been checked and updated throughout the text.
- Mechanism of microbial action is given for gram-negative bacteria alone, adding an illustration for gram-positive bacteria too is required.
R: Thank you for this comment which helps us improve the quality of the manuscript. The mechanism of action is identical; therefore, we prefer not to added another illustration. Otherwise, we have modified the image by illustrating in its corner the generic cell walls structure of Gram positive and negative bacteria and mentioning the different structural complexity between the two.
- Graphical representations, illustrations, and tables for the obtained data can be more meaningful to the readers.
R: More tables have been added in accordance to your suggestions.
- The Discussion part is poorly written. So, the authors should enrich this section with recent relevant literature.
R: Thank you for this important advice. Another reviewer also advised us to improve this part. In accordance with both suggestions, we have created a new paragraph, entitled 'limitations and future perspectives', in which there is more in-depth reasoning regarding the implications of using EOs as antimicrobials and more up-to-date literature has been cited.
- In the table authors have mentioned family names for some species alone, try to include for all the species.
R: Unfortunately, we cannot include this information for all species. Often, the authors of the aforementioned study did not make a detailed identification of the botanical species assayed but limited themselves to indicating only the genus.
- Line 65 Justify the sentence.
R: Amended.
- Lines 119, 123-126, 128-130, 214, 297-298, 306-307, 311-313, 325, 607, 618-619, 631, 655-657, 662, 712, 734 Rephrase all these sentences without grammatical errors.
R: Thanks for your advice. The sentences have been rephrased and corrected.
- 131 family
R: now amended.
- Lines 230 and 235 specific epithets must be in lower case.
R: now amended.
- Line 246 what does contaminating eggs represent the sentence?
R: The sentence was written to better express the concept.
- Line 289 how E. colicould show inhibition?
R: The sentence was written to better express the concept.
- Line 316 check the binomial name for citrus lemon.
R: now amended.
- Line 317 check the spelling mistake in binomial and author citation Zingiber officinalis Rosc.
R: now amended.
- Line 319 scientific name must be in italics
R: now amended.
- Line 337 check the spelling of Salmonella and italics.
R: now amended.
- Line 338 Check the binomial name ‘S. abortive ovis’
R: now amended.
- Line 340 full stop missing after genus abbreviation ‘Spullorum’
R: now amended.
- Lines 353, 354 check the scientific names and italics it.
R: now amended.
- Line 359 Uppercase in between sentence must be avoided
R: now amended.
- Combine the sentence 387 and 388 with relevant paragrapgh
R: The phrase has been moved to the discussion.
- What is that term RCMB in lines 417, 425
R: The acronym, in the original article, stands for Regional Center for Mycology and Biotechnology. It accompanies and precedes the ATCC. It has now been removed in the review to avoid misunderstandings.
- Check the author citation for the binomial line 431
R: Checked. Now it’s correct.
- Line 448 Write the binomial with author citation and check its current name.
- Line 489 italics for the scientific name is required.
R: now amended.
- 521, 527 full stop is missing between abbreviated genus and species name.
R: now amended.
- Line 528 and 529 mention the part from which essential oil is extracted and the sentence must be rephrased.
R: in the original article it is not specify the part from which the EO was isolated. To better explain the concept, the sentence has been rephrased as suggested.
- Line 544 Et al. isn’t plural already the word itself is self-explanatory.
R: now amended.
- Line 555 author shouldn’t include the year of publication for the references cited in the running text according to author guidelines. Stick to the pattern of the journal.
R: now amended.
- Line 600 check the spelling for the bacterial strain
R: now amended.
- Line 618, 619 what does the authors mean by the usage of word ‘just’?
R: deleted.
- Line 649 species name must not start with uppercase
R: now amended.
- Line 662 Usage of ATCC is irrelevant, rephrase the sentence
R: modified as suggested.
- Line 677 include the concentration
- Lines 755, 759 M. avium aviumspecies name is repeated
R: now amended.
- Line 767 Denote the term sub species
R: Now amended.
Reference style in running text must be checked with author guidelines and change it in the required places.

Reviewer 3 Report
Comments and Suggestions for Authors
The choice of the topic is extremely timely, since the emergence of antibiotic resistance is a worldwide problem in both human and veterinary medicine.
The manuscript is extremely well structured, but I have a few critical comments.
In my opinion, the manuscript should cover how the antibacterial effect of essential oils can be used in practice. Essential oils can be irritating and toxic, and have an extremely strong antibacterial effect. In my opinion, the use of essential oils against pathogens causing problems in veterinary medicine, is limited.
My further critical observation is that in table 1, the auctor name belonging to the scientific name of the plants only appears in some places, and the family appears in some places. This must be unified. Also, as long as the scientific name of the given plant is listed, it will last for a short time.
After revising the manuscript.
Comments on the Quality of English LanguageThe English of the article is correct, it contains some typos.
Author Response
The choice of the topic is extremely timely, since the emergence of antibiotic resistance is a worldwide problem in both human and veterinary medicine.
The manuscript is extremely well structured, but I have a few critical comments.
R: Thank you for your appreciation and for the time you took to review the manuscript. We have made the required changes which you can find highlighted.
In my opinion, the manuscript should cover how the antibacterial effect of essential oils can be used in practice. Essential oils can be irritating and toxic, and have an extremely strong antibacterial effect. In my opinion, the use of essential oils against pathogens causing problems in veterinary medicine, is limited.
R: Thank you for this important advice. A new paragraph describing the limits and potential related to the use of essential oils in veterinary medicine has been added to the text. The new paragraph is entitled: Limitations and future perspective
My further critical observation is that in table 1, the auctor name belonging to the scientific name of the plants only appears in some places, and the family appears in some places. This must be unified. Also, as long as the scientific name of the given plant is listed, it will last for a short time.
R: We modified the table by deleting additional and inconsistent information. Concerning your point about the scientific name, we think it is more appropriate to keep this rather than giving the common name. The properties of the same plant species may vary in relation to different conditions (insolation, soil composition, cultivation methods, extraction methods, etc.), including in relation to the specific subspecies or variety.
After revising the manuscript. The English of the article is correct, it contains some typos.
R: The manuscript has been double-checked entirely for typos

Reviewer 4 Report
Comments and Suggestions for Authors
The manuscript antibiotics-2832421 is a review about the veterinary use of essential oils against some bacteria species.
major comments: The authors must justified the assertion L111 "There is a lack of literature reviews in this field." Contrary, in conclusion section L858 "A simple search using the PubMed database reveals hundreds of publications in the scientific and medical literature describing the antibacterial properties of various plant species and their chemical components." For clarity, please rephrase the aim of the review.
The authors should provide the workflow of their review - e.g. PRISMA. In order to minimize the bias there are some methods for selecting the references included in the review. Even a literature review needs a methodology. What types of publications are selected - reviews, research articles, conference presentations, book chapters, etc. ? What keywords are used for searching the literatures in the public databases? What databases were interrogated? What period of time was investigated? etc.
L226 - Enterobacteriaceae is replace by Enterobacterales.
minor comments: Some sentences are confused, e.g. L32 "the treatment of the main bacterial genera of veterinary interest in livestock." We treat the bacterial infections not bacteria genera.
L57-58 "In 2014, the World Health Organization (WHO) stated that the global public health catastrophe posed by antibiotic-resistant bacteria was approaching quickly [8]." In my opinion, it's 2024 and there are other global public health catastrophes. Of course, multi-drug resistant strains are evolving and spreading rapidly, which justifies the concerns of healthcare professionals.
Author Response
The manuscript antibiotics-2832421 is a review about the veterinary use of essential oils against some bacteria species.
major comments: The authors must justified the assertion L111 "There is a lack of literature reviews in this field." Contrary, in conclusion section L858 "A simple search using the PubMed database reveals hundreds of publications in the scientific and medical literature describing the antibacterial properties of various plant species and their chemical components." For clarity, please rephrase the aim of the review.
R: Thank you for your important revision work on our manuscript. In accordance with your suggestion, we have rewritten the sentence introducing the objectives of the review, making the concept clearer.
The authors should provide the workflow of their review - e.g. PRISMA. In order to minimize the bias there are some methods for selecting the references included in the review. Even a literature review needs a methodology. What types of publications are selected - reviews, research articles, conference presentations, book chapters, etc. ? What keywords are used for searching the literatures in the public databases? What databases were interrogated? What period of time was investigated? etc.
R: A specific paragraph, entitled “Methodology”, has been added to the text. In it, the methods carried out in writing the review have been detailed.
L226 - Enterobacteriaceae is replace by Enterobacterales.
R: now amended.
minor comments: Some sentences are confused, e.g. L32 "the treatment of the main bacterial genera of veterinary interest in livestock." We treat the bacterial infections not bacteria genera.
R: Many thanks for this advice. The sentence was rewritten to make the concept clearer.
L57-58 "In 2014, the World Health Organization (WHO) stated that the global public health catastrophe posed by antibiotic-resistant bacteria was approaching quickly [8]." In my opinion, it's 2024 and there are other global public health catastrophes. Of course, multi-drug resistant strains are evolving and spreading rapidly, which justifies the concerns of healthcare professionals.
R: the report is from 2014 as can be seen from the bibliographic citation. Like you, we are aware that there are other important adversities facing the world of scientific research. However, we believe that antimicrobial resistance is equally important. Therefore, also depending on the type of review we have written, we have taken care to report this important consideration.
The manuscript antibiotics-2832421 is a review about the veterinary use of essential oils against some bacteria species.
major comments: The authors must justified the assertion L111 "There is a lack of literature reviews in this field." Contrary, in conclusion section L858 "A simple search using the PubMed database reveals hundreds of publications in the scientific and medical literature describing the antibacterial properties of various plant species and their chemical components." For clarity, please rephrase the aim of the review.
R: Thank you for your important revision work on our manuscript. In accordance with your suggestion, we have rewritten the sentence introducing the objectives of the review, making the concept clearer.
The authors should provide the workflow of their review - e.g. PRISMA. In order to minimize the bias there are some methods for selecting the references included in the review. Even a literature review needs a methodology. What types of publications are selected - reviews, research articles, conference presentations, book chapters, etc. ? What keywords are used for searching the literatures in the public databases? What databases were interrogated? What period of time was investigated? etc.
R: A specific paragraph, entitled “Methodology”, has been added to the text. In it, the methods carried out in writing the review have been detailed.
L226 - Enterobacteriaceae is replace by Enterobacterales.
R: now amended.
minor comments: Some sentences are confused, e.g. L32 "the treatment of the main bacterial genera of veterinary interest in livestock." We treat the bacterial infections not bacteria genera.
R: Many thanks for this advice. The sentence was rewritten to make the concept clearer.
L57-58 "In 2014, the World Health Organization (WHO) stated that the global public health catastrophe posed by antibiotic-resistant bacteria was approaching quickly [8]." In my opinion, it's 2024 and there are other global public health catastrophes. Of course, multi-drug resistant strains are evolving and spreading rapidly, which justifies the concerns of healthcare professionals.
R: the report is from 2014 as can be seen from the bibliographic citation. Like you, we are aware that there are other important adversities facing the world of scientific research. However, we believe that antimicrobial resistance is equally important. Therefore, also depending on the type of review we have written, we have taken care to report this important consideration.

Round 2
Reviewer 1 Report
Comments and Suggestions for Authors
The manuscript is sufficiently improved and may be accepted in its present form for its possible publication in Antibiotics.
Reviewer 2 Report
Comments and Suggestions for Authors
I wish to express my sincere appreciation to the authors for their revised manuscript, diligent focus on details, and adept handling of the issues raised during the initial review. Their efforts have resulted in substantial improvements, significantly elevating the quality and clarity of the manuscript. It is suggested to authors that, while responding to the reviewers and editorial comments, the authors should mention the page number/line number for the amendments made by them for a clear understanding of the revision.
Reviewer 4 Report
Comments and Suggestions for Authors
The authors responded the comments that I addressed.
If the editors consider that the manuscript meets the Antibiotics's requirements, then it can be published.